# Cellulose in Secondary Xylem of Cactaceae: Crystalline Composition and Anatomical Distribution

**DOI:** 10.3390/polym14224840

**Published:** 2022-11-10

**Authors:** Agustín Maceda, Marcos Soto-Hernández, Teresa Terrazas

**Affiliations:** 1Instituto de Biología, Universidad Nacional Autónoma de México, Mexico City 04510, Mexico; 2Programa de Botánica, Colegio de Postgraduados, Montecillo 56230, Mexico

**Keywords:** crystalline cellulose, FTIR, fluorescence microscopy, crystallinity indexes, Cactaceae

## Abstract

Cellulose is the main polymer that gives strength to the cell wall and is located in the primary and secondary cell walls of plants. In Cactaceae, there are no studies on the composition of cellulose. The objective of this work was to analyze the crystallinity composition and anatomical distribution of cellulose in Cactaceae vascular tissue. Twenty-five species of Cactaceae were collected, dried, and milled. Cellulose was purified and analyzed with Fourier transform infrared spectroscopy, the crystallinity indexes were calculated, and statistical analyzes were performed. Stem sections were fixed, cut, and stained with safranin O/fast green, for observation with epifluorescence microscopy. The crystalline cellulose ratios had statistical differences between *Echinocereus pectinatus* and *Coryphantha pallida*. All cacti species presented a higher proportion of crystalline cellulose. The fluorescence emission of the cellulose was red in color and distributed in the primary wall of non-fibrous species; while in the fibrous species, the distribution was in the pits. The high percentages of crystalline cellulose may be related to its distribution in the non-lignified parenchyma and primary walls of tracheary elements with helical or annular thickenings of non-fibrous species, possibly offering structural rigidity and forming part of the defense system against pathogens.

## 1. Introduction

Plant cell walls give rigidity, delimit the cell space, and function as a physical barrier against pathogens [1]. The main polymers of the cell wall are cellulose, hemicelluloses, and lignin, in addition to other compounds such as pectins, structural proteins, and glycoproteins [2]. The cell wall is divided in two: the primary wall with abundant cellulose microfibers, some hemicelluloses, xyloglucans, and pectins [3]. The primary wall develops during cell growth and maintains some elasticity during the initial stage. The secondary wall has three layers: S1, S2, and S3. A secondary wall develops once cell growth ends, mainly lignin accumulates, and cellulose occurs in a lesser quantity than other polysaccharides such as hemicelluloses [4].

Cellulose is the main component of the cell wall [1] and is a homopolysaccharide composed of repeated glucose residues linked by β-(1→4) bonds that generate long and rigid microfibrils [5] made of 18 cellulose polymers [6]. The presence of cellulose microfibrils in the cell wall confers mechanical and enzymatic resistance [7], structural rigidity in the primary wall [2], and forms the scaffolding for binding with pectins and hemicelluloses [5]. Two types of cellulose are characterized by their orientation and packaging. The cellulose formed by microfibrils is also called crystalline cellulose [8], which is packaged because the microfibrils are linked by hydrogen β-(1→4)-linked D-glucose units, making it more compact, rigid, and ordered [9]. The cellulose matrix that has no order (amorphous) forms a network where the hemicelluloses, pectins, lignin, and phenolic compounds are inserted and joined [3,10].

The importance of studying the crystalline composition of cellulose is due to its use in the paper industry [8] and for the production of biofuels [11]. In addition, Cactaceae species are considered second generation plants for biofuel production because, despite their slow growth, they resist drought conditions and high temperatures, and are not an essential part of human consumption [12]. The species with the highest amount of amorphous cellulose [13] and the lowest presence of lignin or lignin, and with the highest accumulation of syringyl monomers [14], are those that have potential for use in the previously mentioned industries. On the other hand, cellulose studies also focus on anatomical-structural analysis, to identify cell wall interactions with biotic and abiotic factors [15]. At this point, different authors have analyzed the stress effects of the environment on cellulose accumulation [16] and the interaction between cellulose and pathogens [17].

In succulent species, studies are scarce and most have focused on the genus *Agave*, a member of the Asparagaceae family [18]. In the Cactaceae family, few studies have identified the chemical composition of lignin [19,20,21,22]; furthermore, the crystallinity of cellulose was only analyzed in *Opuntia ficus*-*indica,* due to the potential use of its cellulosic compounds in the biofuel and paper industry [23]. However, of the other species of the Opuntioideae, Pereskioideae, and Cactoideae subfamilies, there have been no studies on the composition of cellulose and its distribution in the secondary xylem (wood), which would allow us to understand the structure and functioning of cellulose in the cell wall of the parenchyma, water conductive, and supporting cells. Therefore, this study aimed to analyze the crystalline composition of cellulose and its distribution in wood. The hypothesis was that there would be variation in the crystalline composition of cellulose among the different groups of cacti, due to the type of wood present.

## 2. Materials and Methods

### 2.1. Crystalline Analysis of Cellulose

To analyze the crystalline cellulose of Cactaceae species, adult and healthy plants of representative species of Cactaceae (Table 1) were collected. The vascular cylinder of all species was isolated and dried for 72 h. After that, 2 g of each species was weighed, dried, and ground. Subsequently, to obtain free-extractive wood, successive extractions were applied with ethanol: benzene (1:2), ethanol 96%, and hot water at 90 °C, according to the method proposed by Reyes-Rivera and Terrazas [24].

For the free-extractive wood, cellulose purification was performed using the procedure of Maceda et al. [20] based on the Kürschner-Hoffer method. Whereby, 0.5 g of extractive-free wood was weighed and added to 12.5 mL of HNO3/ethanol (1:4 *v*/*v*), kept in a reflux system for 1 h, allowed to cool to room temperature, and the sample was decanted. Subsequently, 12.5 mL of HNO_3_/ethanol was added and the process of reflux, cooling, and decantation was repeated two more times. In the last process, 12.5 mL of 1% aqueous KOH solution was added, kept under reflux for an additional 30 min, and finally filtered through a fine-pore Büchner filter. The residue (cellulose) was dried at 60 °C for 12 h.

Obtainment of the crystalline and amorphous cellulose proportion was performed with attenuated total reflection Fourier transform infrared spectroscopy (ATR-FTIR), because the covalent and non-covalent interactions of cellulose could be identified, and the crystallinity of cellulose could be measured [25]. A small amount of dry cellulose from each species was taken, readings in the spectrum range of 4000–650 cm^−1^ (30 scans with a resolution of 4 cm^−1^, 15 s per sample) of each sample were made in an FTIR Spectrometer (Agilent Cary 630 FTIR), and a baseline correction was made with MicroLab PC software (Agilent Technologies, Santa Clara, CA, USA). The raw spectra were converted from transmittance to absorbance, and the average of three spectra was obtained using the Resolution Pro FTIR Software program (Agilent Technologies, Santa Clara, CA, USA).

The total crystallinity index (TCI) proposed by Nelson and O’Connor [26], or also called the proportion of crystallinity [13], was calculated from the ratio between the intensity absorption peaks 1370 cm^−1^ and 2900 cm^−1^ [27]. The lateral order index (LOI) [26,27], or the second proportion of crystallinity [13], was calculated from the ratio between the intensity absorption peaks 1430 cm^−1^ and 893 cm^−1^. Hydrogen bonding intensity (HBI) was calculated from the ratio between 3350 cm^−1^ and 1315 cm^−1^ [28].

The data obtained in triplicate for TCI, LOI, and HBI for each species were analyzed with non-parametric statistics, due to the data having no normality with the Kolmogorov-Smirnov and Shapiro–Wilk tests, even with the square root and log transformations. A non-parametric Kruskal–Wallis test was used to determine if there were differences among the species, followed by Dunn’s post hoc test.

### 2.2. Epifluorescence Microscopy

To analyze the anatomical distribution of cellulose in Cactaceae species, epifluorescent microscopy was used. Epifluorescence is a technique used to identify the distribution of structural components such as lignin, cellulose, phenolic compounds, starch, and proteins [29]. In the particular case of epifluorescence, it allows the observation of thin sections of plant samples, where the tissues are anisotropic [30]. The method used to make the observations was based on the results obtained by De Micco and Aronne [31] and Maceda et al. [21,22] with epifluorescence and safranin O/fast green staining (SF). However, to confirm that the fluorescence emission by safranin/fast green staining corresponded to the fluorescence of lignified tissues and cellulose, samples of *Ferocactus latispinus* (HAW.) Britton and Rose were used as a model for comparison with two other typical stains for fluorescence, such as acridine orange–Congo red (AO), in addition to calcofluor (CA), based on the method proposed by Nakaba et al. [32].

Representative samples of wood from the base of the stem of the Cactaceae species were obtained, fixed with a solution of formalin–acetic acid–ethanol (10:5:85) [33], and washed, before dehydrating the samples [30]. The non-fibrous species samples were embedded with paraffin and cut with a rotatory microtome, and the fibrous species were cut with a sliding microtome [20]. All the samples were stained with safranin O/fast green and mounted in synthetic resin according to Loza-Cornejo and Terrazas [34].

Due to the different structural components of the cell wall, presenting autofluorescence and fluorescence emission [29], three different excitation and emission bands were used at the same time [21]. This process allowed obtaining images with a “true color” [29]. Therefore, a wide-field fluorescence microscope (Zeiss Axio Imager Z2) with Apotome 2.0 (Zeiss Apotome.2), an AxioCam MRc 5 (Zeiss), and a microscope metal halide fluorescence light source (Zeiss HXP 120) were used. The multicolor images were obtained with a triple excitation/emission bandwidth: DAPI (blue) with an excitation of 365 nm and emission of 445/50 nm, FITC (green) with an excitation of 470/40 nm and emission of 525/50 nm, and TRITC (red) with an excitation of 546/12 nm and emission of 575–640 nm. Each sample was exposed to fluorescence light with a low power for a maximum of one minute, as proposed by Baldacci-Cresp et al. [35], to avoid overexposure of the samples and cause their photobleaching. Images were obtained with the Zen Blue 2.5 lite (Zeiss, Jena, Germany) program, and brightness adjustments were made to the entire image.

## 3. Results

### 3.1. Cellulose Composition

Figure 1 illustrates the cellulose spectra and Table 2 shows the allocation of the main bands. The absence or presence of small bands at 1269 cm^−1^ corresponding to lignin and hemicelluloses; 1595 cm^−1^, 1512 cm^−1^, and 1463 cm^−1^ assigned to lignin; and 1735 cm^−1^ corresponding to xylenes and hemicelluloses reflected the effectiveness of the extraction and cellulose purification. The presence of 1640 cm^−1^ reflected the O-H vibration of absorbed water. The parameters for crystallinity, TCI, LOI, and HBI, had statistically significant differences (*p* < 0.05) with a non-parametric Kruskal–Wallis analysis (Table 3); and using Dunn’s post hoc test, the species that were statistically different were identified (Table 4 and the website on Data Availability Statement contain the tables for each variable and the comparison between species).

The TCI showed that all species had values above one, because cacti have a higher proportion of crystalline cellulose than amorphous cellulose (Table 4). *E. pectinatus* had the highest TCI values and had significant differences with *C. pallida* and *C. clavata*, which presented the lowest values (Table 4). The lateral order index (LOI) reflected the degree of order in cellulose and the presence of crystalline cellulose II or amorphous cellulose; all species had similar values, except for *E. pectinatus, C. pallida,* and *C. ramillosa,* because the last two presented significant differences from the first (Table 4). In HBI, which relates to the crystal system and the bound water, the significant differences were between *C. pallida* with the highest values, and *E. pectinatus*, which presented the lowest HBI of the cacti (Table 4). The crystallinity indexes TCI and HBI were high in all Cactaceae species, but had lower values in the LOI index, because the cacti species had a lesser order (disorder) in the crystalline structure, but a higher proportion of crystalline cellulose.

### 3.2. Staining Methods for Fluorescence

When comparing the bright-field images with the fluorescence images, it was observed that in the bright-field images, the blue tones corresponded to the non-lignified tissue, while the red tones corresponded to the lignified tissue. In the fluorescence images, the green tones reflected the lignified tissue and the red the non-lignified (Figure 2A,B). In the SF, AO, and CA stains (Figure 2) it was observed that the secondary walls of the vessel elements (v) and wide-band tracheids (wbt) showed fluorescence emission in green to bluish tones in all three types of staining. In non-lignified tissues, such as the parenchyma or the primary walls of the vascular tissue, in the SF and AO stains, the cellulose fluoresced in red tones, while, for CA, the tones were bluish to greenish (Figure 2D).

### 3.3. Cellulose Distribution in Cells of the Secondary Xylem

Figure 3, Figure 4, Figure 5 and Figure 6 show representative species of fibrous (Pereskioideae and Opuntioideae) and non-fibrous (Cactoideae) wood. In the bright-field microscopy images, the lignified cell walls were observed in red tones, with the non-lignified in blue tones. With fluorescence microscopy, the fibrous species reflected bluish-green to yellow tones; while in the non-fibrous species, the tones of the lignified walls were greenish. The presence of cellulose and other components of the non-lignified walls reflected reddish tones in all species. *Cylindropuntia leptocaulis* (Figure 3A,B) had starch within some parenchyma (p) cells, which were cyan-colored. The distribution of cellulose was different between the fibrous species and non-fibrous species.

In longitudinal sections of *Cylindropuntia*, *Opuntia*, and *L. lychnidiflora,* cellulose was detected in the pits of the VEs (Figure 4A,B,D). In addition, the presence of cellulose was mainly in the simple pits of the fibers (f) of *Cylindropuntia* and *L. lychnidiflora*. (Figure 4A,B,D). The unlignified parenchyma of *O. stenopetala* in longitudinal sections showed primary walls with the presence of cellulose (Figure 4C); whereas, in the lignified parenchyma of *Cylindropuntia* and *Leuenbergeria*, lignin was mainly accumulated in the cell wall, and cellulose was exclusively in the pits (Figure 4A,B,D).

In non-fibrous species (Figure 5), cellulose fluorescence emission was detected in the primary wall of p, wbt, and v; unlike in fibrous species, where fluorescence emission was not detected in the primary wall but in the S3 layer of the secondary wall adjacent to the lumen. Furthermore, the non-fibrous species showed secondary walls as helical and annular thickenings in v and wbt, so the primary wall was visible in the longitudinal sections (Figure 6). In all non-fibrous species, the p was abundant and unlignified, in addition to having a greater accumulation of cellulose (Figure 5B,D,F,H and Figure 6B).

## 4. Discussion

All Cactaceae species had a higher cellulose crystalline proportion, and the distribution of cellulose was in unlignified parenchyma, v, and wbt; all with secondary walls as helical and annular thickenings.

### 4.1. Crystalline Indexes

The TCI index was based on the proportion between the intensity peaks of 1370 cm^−1^ and 2900 cm^−1^, according to Nelson and O’Conner [26]. The TCI is proportional to the degree of crystallinity of cellulose; therefore, the higher the ratio, the higher the percentage of crystalline cellulose, as reported by various authors for different species, fibers, and materials [27,28,38]. In the case of the LOI index, the peak of 1430 cm^−1^ corresponds to the presence of crystalline cellulose I, while the peak of 893 cm^−1^ reflects the presence of crystalline cellulose II and amorphous cellulose [39]. LOI reflects the ordered regions perpendicular to the chain direction, which is influenced by the chemical extraction and purification of cellulose [40]. The low values of LOI for Cactaceae species could have been the result of the presence of crystalline cellulose II, which influences the peak 893 cm^−1^ [41], or the effect of the temperature and concentration of the NaOH solution during the purification of cellulose [39]. In the case of HBI, these values are useful for interpreting qualitative changes in cellulose crystallinity; the lower values indicate the presence of crystalline cellulose, and if the values are higher, this could indicate the presence of cellulose II or amorphous cellulose. However, these values could represent the amount of bound water in the fiber structure and the presence of extractives, hemicelluloses, and lignin that increase HBI values [27].

The results observed in Table 4 reflect that all cacti species had high proportions of crystalline cellulose, because they had high values of TCI and LOI, and similar values in HBI. The species *E*. *pectinatus*, *C. retusa*, and *C. ramillosa* (except in the LOI value) had the highest crystalline proportions, due to having the highest values of TCI and LOI, and lower values of HBI, which corresponds with the fibers and materials with high crystalline cellulose proportions reported by Colom and Carrillo [42], Carrillo et al. [43], Široký et al. [39], and Poletto et al. [27].

Comparing with the reports in the literature for other cacti species, the high proportions of crystallinity cellulose were similar to most angiosperms reported by Agarwal et al. [44] and to the bark of the cactus *Cereus forbesii*, which had a percentage of 82% crystalline cellulose [45]. *Opuntia ficus*-*indica* is a species with varying crystalline cellulose percentages in different parts of the plant: cladodes 27% [23] and 79% [46], spines 34–70% [47,48], vascular tissue 22–28% [49], fruit epidermis 38% [50], and 60% in seeds [51]. Maceda et al. [46] reported percentages of crystalline cellulose of 76% and 74%, respectively, for *Opuntia streptacantha* and *O. robusta.* These values are very similar to those reported in this study for non-fibrous and fibrous species. Therefore, high proportions or percentages of crystalline cellulose is a constant in different Cactaceae species, probably due to the presence of non-lignified parenchyma or non-lignified primary cell walls, as observed in the anatomical distribution. Further studies comparing the purity of cellulose using the Kürschner-Hoffer method with other methods, such as the Seifert [52] method, would allow corroborating the proportions of crystalline cellulose in the Cactaceae species analyzed.

### 4.2. Staining Methods for Fluorescence

With the three staining methods (SF, AO, and CA; Figure 2), the distribution of the lignified tissue from the non-lignified was identified. The tones in the fluorescence emission for lignin in SF were similar to what was observed in AO and CA, and to what was reported by other authors [22,29,30]. The use of SF for bright-field and fluorescence is advantageous over AO and CA, because images can only be taken using fluorescence microscopy. On the other hand, the AO and CA techniques are semi-permanent, so over time the fluorescence is lost, when the dyes become diluted in the mounting medium used (phosphate buffer:glycerol 1:1 *v*/*v*) [30]; while for SF, a permanent mounting medium is used, so slides can be stored without loss of fluorescence.

Finally, the difference in lignin tones was visible in SF and CA, such as v, whose intensity and fluorescence tonalities were different from wbt; while for AO, the tonalities were similar in v and wbt. For the distribution of cellulose in the non-lignified parenchyma and in the vascular tissue, SF was clearly observed in red tones; while in AO and CA, the fluorescence emission of the parenchyma was obscured by the fluorescence of the vascular tissue. Therefore, SF was an efficient method for determining the distribution of lignified tissues and cellulose, as was previously reported in the literature [21,22].

### 4.3. Cellulose Distribution and Crystalline Composition

The tonalities observed in the fluorescence emission for cellulose and lignin correspond to those reported in other similar studies with a safranin O/fast green staining technique and three excitation bands [21,22,29,30,31]. This technique allowed the detection of differences in the distribution of both structural compounds (cellulose and lignin), between non-fibrous and fibrous species. The safranin O dye allows the detection of lignin autofluorescence and fluorescence in analyzed tissues [30,31], with tonalities of blue to green [29,35]. In the case of celluloses and hemicelluloses that do not have autofluorescence, such as lignin [29], with safranin O/fast green staining, the fluorescence emission of cellulose can be detected at 570–620 nm and its tonalities were reddish, similarly to the results of Maceda et al. [21,22]. However, further studies with immunohistochemistry or specific fluorophores could help confirm the distribution of cellulose [2,29].

When comparing the results with those reported by Maceda et al. [21] for the primary xylem, cellulose accumulated in the primary walls and lignin in the secondary ones of the helical and annular thickenings of the protoxylem and metaxylem of fibrous species, similarly to what was observed in non-fibrous adult plants. Only in the metaxylem of *Leuenbergeria lychnidiflora* was there a decrease in the accumulation of cellulose in the primary walls and a greater accumulation in the intervascular pits, as obtained in the secondary xylem of the fibrous species.

The presence of high proportions of crystalline cellulose in all studied cacti, mainly in the non-fibrous species (*E. pectinatus*, *C. retusa,* and *C. ramillosa*), could be related to the abundant unlignified parenchyma and the lower accumulation of lignin in the xylem, as seen in Figure 4 and Figure 5B. Even when including fibrous species of *Cylindropuntia* and *Opuntia*, there was a high proportion of crystalline cellulose, similarly to non-fibrous species, possibly because the xylem had a greater accumulation of lignin in the cell walls of the v, f, and p [20]. This high proportion of lignin may function as a physical [53] and chemical barrier against the attack of pathogens [54]. Zhao and Dixon [7] and Bacete and Hamann [55] mentioned that the cell wall is a dynamic barrier in conditions of abiotic stress. Therefore, in some cells, the presence of increased lignin accumulation can be observed [56]; while in others, similarly to in the gelatinous layer (G), cellulose is mainly accumulated [8].

In the cells with non-lignified primary cell walls, the accumulation of crystalline cellulose packed by hydrogen chain β-(1→4)-linked D-glucose units [9] makes cellulose more hydrophobic than amorphous cellulose [8], conferring structural support [9] and causing a decrease in the efficiency of cellulase enzymes [57], by not presenting sites for binding with enzymes and making it difficult to degrade [58]. In contrast, amorphous cellulose is slightly hydrophilic [55], susceptible, and degrades rapidly, due to the action of cellulase enzymes [59] and pH changes from pathogens [60].

Infection with some pathogenic fungi occurs when the hyphae invade the roots and subsequently the vascular tissue [16]. In the fibrous species of Cactaceae, the accumulation of lignin works as a physical barrier in Vs [61]. However, in non-fibrous species with little accumulation of lignin in the helical or annular thickenings, the presence of crystalline cellulose in the primary wall of the tracheary elements and the unlignified parenchyma function as a physical barrier, to prevent the spread of the fungus by reducing the effectiveness of cellulase enzymes [55]. For the species analyzed in this work, it has been reported that they have higher percentages of cellulose than lignin in the vascular tissue [20]; thus, possibly, the presence of crystalline cellulose reinforces and protects the vessels with helical and annular secondary thickenings. In subsequent studies using transmission electron microscopy techniques [62], the presence of crystalline cellulose in the primary wall of the tracheary elements could be analyzed and characterized, which will support and confirm this assertion.

The presence of high proportions of crystalline cellulose in fiber species could be due to the succulence of their stems and their distribution in humid environments, such as for *Leuenbergeria lychnidiflora*, [63] or in the extreme conditions of arid and semi-arid climates with seasons of high humidity. The resistance of plants to stressful conditions is energetically expensive, in addition to the constitutive expression of defense mechanisms, such as the accumulation of callose [1,5], pectins [55], or secondary metabolites [64]. This is not always the best strategy against the colonization of pathogens or diseases, because it can restrict physiological processes and have negative impacts on the plant, such as a reduced seed production and biomass [14]. Therefore, the presence of primary physical barriers, such as lignin [65,66] and crystalline cellulose [9] that inhibit the spread of pathogens, could decrease the expression of the defensive systems (callose, pectins, secondary metabolites) and be energetically expensive [15]. The heterogeneity in the composition of cell walls between species reflects the diversity of defensive mechanisms against the degrading enzymes that pathogens have developed to break down plant cell walls, such as the numerous cell wall-degrading enzymes (CWDEs), polygalacturonases, and xylanases [67].

In these Cactaceae species, as mentioned previously, cellulose may work to confer structural support to the primary wall, without losing flexibility [68], and thus maintaining the cell structure during periods of water stress and rain [69]. Furthermore, the increased amount of crystalline cellulose could function as a defense against pathogens [57], by providing resistance to degradation by glycosyl hydrolase enzymes [9] and enzymes produced by pathogenic fungi [17]. Further analyses in a larger number of dimorphic and non-fibrous cacti, together with other families of succulent plants, may confirm that the presence of crystalline cellulose is due to the presence of non-lignified parenchyma, as was observed in the species analyzed in this work.

In addition, the crystalline cellulose can be used for the production of microfibrillated cellulose nanofibers that could be applied in the production of medical equipment [70,71] or in paper recycling [72]. Amorphous cellulose could be enzymatically degraded for the production of biofuels [23,73]; thus, cacti species have potential for future use [74]. However, it is essential to analyze the profitability in terms of cultivation and plant growth.

## 5. Conclusions

The non-fibrous species presented a high proportion of crystalline cellulose, reflected in their TCI, LOI, and HBI proportions. The distribution of the cellulose was in the primary cell wall of the tracheary elements and the unlignified parenchyma. In fibrous species, the distribution was in the cell wall near the lumen and the simple and alternate pits of vessel elements and fibers. The high proportion of crystalline cellulose could be related to resistance to pathogens, due to the presence of a non-lignified primary cell wall in all the cacti species.

## Figures and Tables

**Figure 1 polymers-14-04840-f001:**
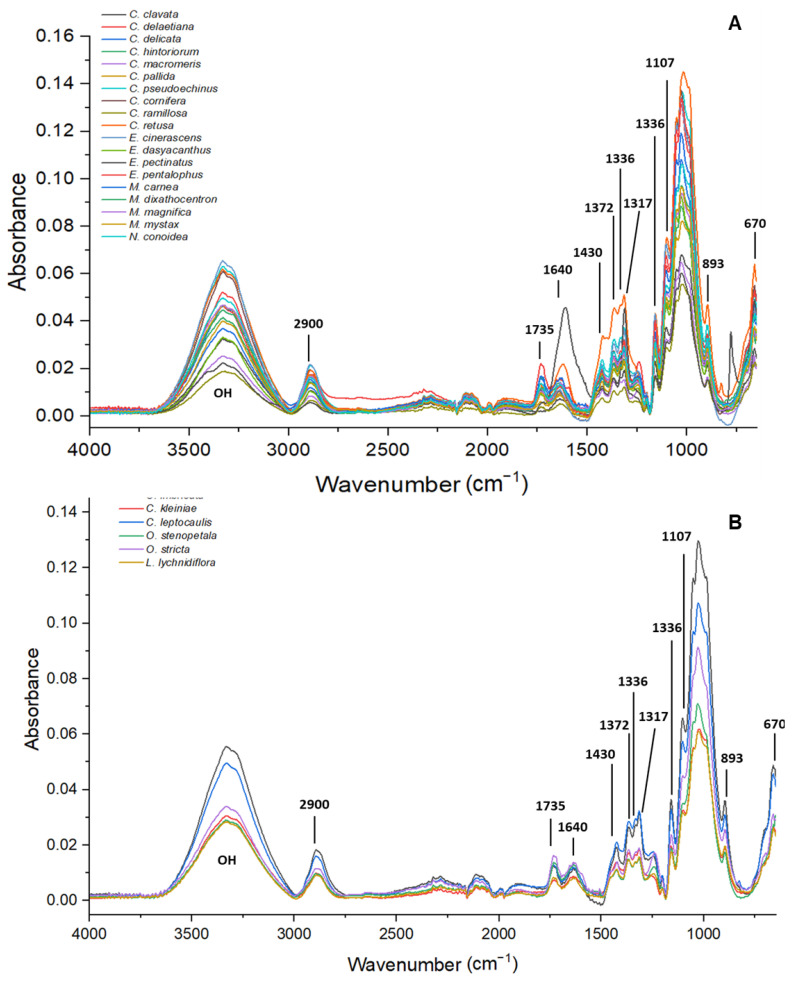
FTIR spectra for obtaining cellulose crystalline indexes for the 25 Cactaceae species. (**A**) Non-fibrous species. (**B**) Fibrous species.

**Figure 2 polymers-14-04840-f002:**
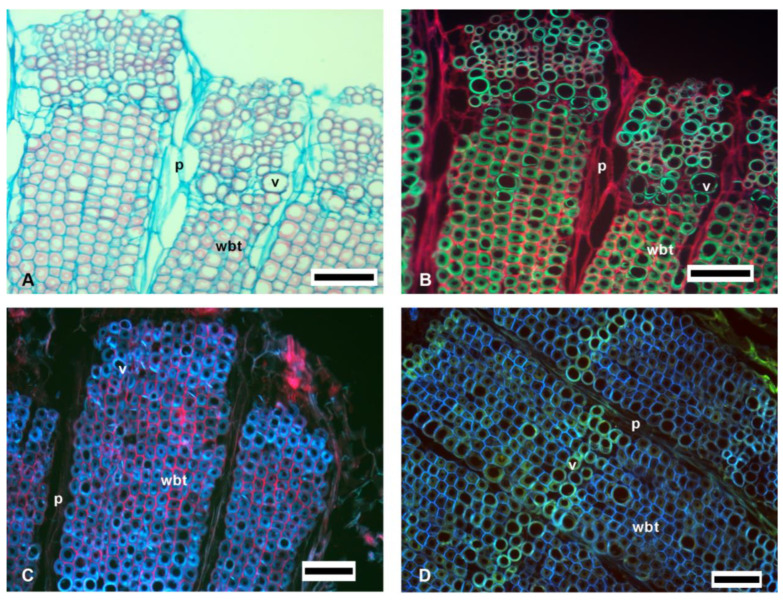
Images of the vascular tissue of *F. latispinus*. (**A**) Bright-field image of SF. (**B**) Fluorescence image of SF. (**C**) Fluorescence image of AO. (**D**) Fluorescence image of CA. p = parenchyma, v = vessel, wbt =wide-band tracheids. Scale: 100 µm.

**Figure 3 polymers-14-04840-f003:**
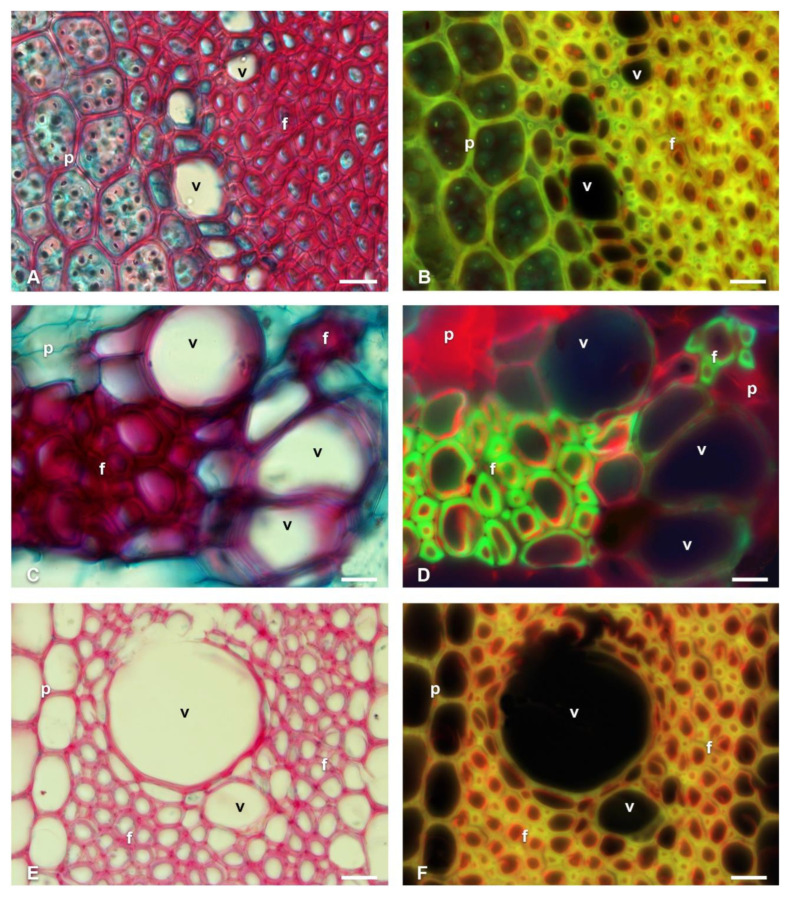
Fluorescence emission of transverse sections of fibrous wood species of Cactaceae. (**A**,**B**) *Cylindropuntia leptocaulis*. (**C**,**D**) *Opuntia stenopetala*. (**E**,**F**) *Leuenbergeria lychnidiflora*. f = fibers, p = parenchyma, v = vessel. Scale: 20 µm.

**Figure 4 polymers-14-04840-f004:**
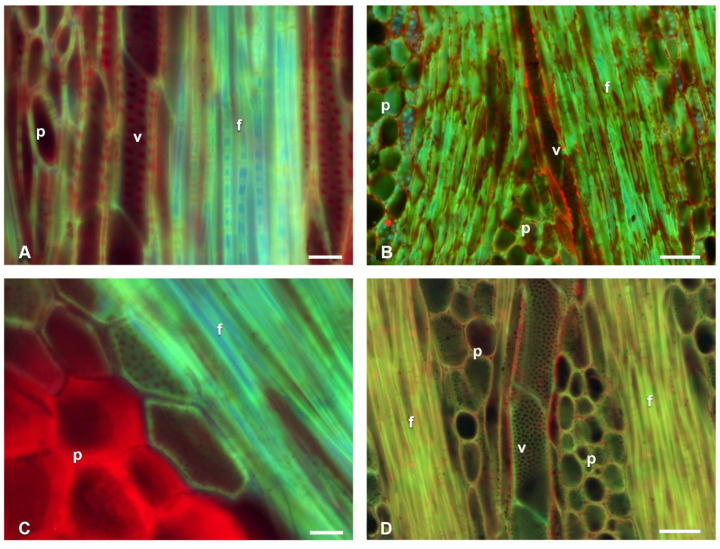
Fluorescence emission of longitudinal sections of fibrous wood species of Cactaceae. (**A**) *Cylindropuntia kleiniae*. (**B**) *Cylindropuntia leptocaulis*. (**C**) *Opuntia stenopetala*. (**D**) *Leuenbergeria lychnidiflora*. f = fibers, p = parenchyma, v = vessel. Scale: 20 µm: (**A**,**C**); 50 µm: (**B**,**D**).

**Figure 5 polymers-14-04840-f005:**
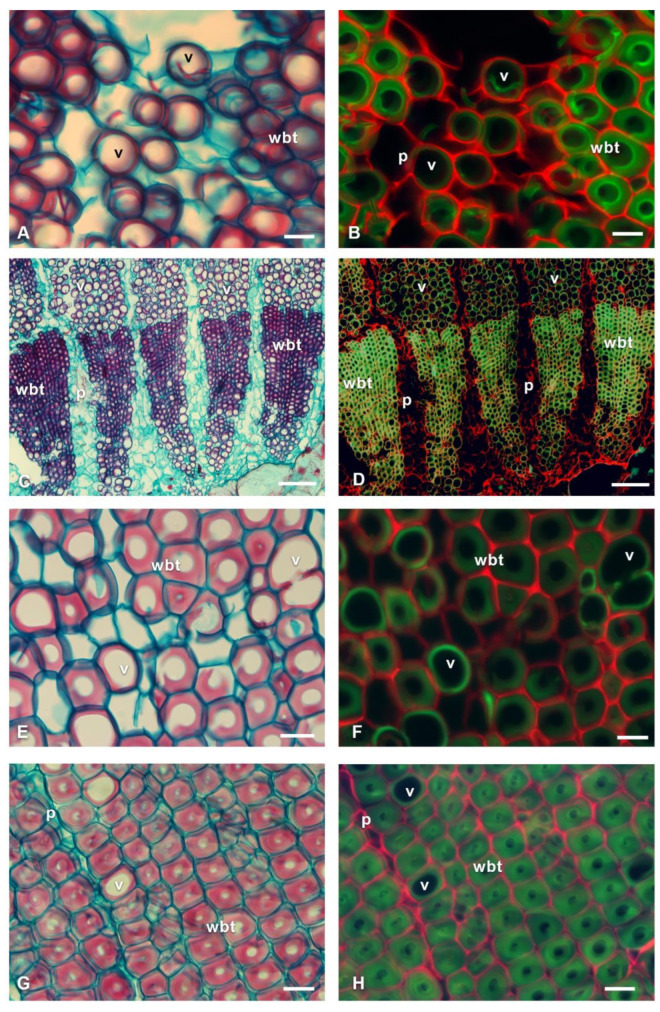
Fluorescence emission of transverse sections of non-fibrous wood species of Cactaceae. (**A**,**B**) *Coryphantha clavata*. (**C**,**D**) *Echinocereus cinerascens*. (**E**,**F**) *Mammillaria dixathocentron*. (**G**,**H**) *Neolloydia conoidea*. p = parenchyma, v = vessel, wbt = wide-band tracheids. Scale: 20 µm: (**A**,**B**,**E**,**H**); 200 µm: (**C**,**D**).

**Figure 6 polymers-14-04840-f006:**
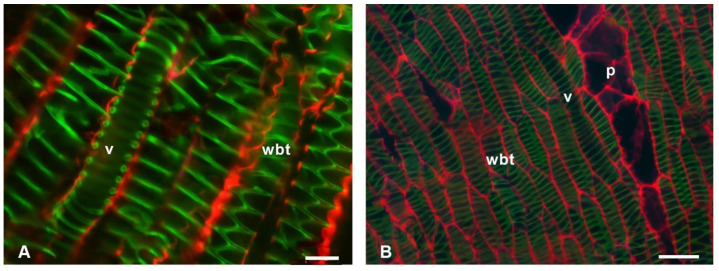
Fluorescence emission of longitudinal sections of non-fibrous wood species of Cactaceae. (**A**) *Carpathia* delaetiana. (**B**) *Neolloydia conoidea*. p = parenchyma, v = vessel, wbt = wide-band tracheids. Scale: 20 µm: (**A**,**B**).

**Table 1 polymers-14-04840-t001:** Morpho-anatomical characteristics of the 25 species of Cactaceae.

Subfamily	Species	Collection Number	Wood Type	Stem
Cactoideae	*Coryphantha clavata* (Scheidw.) Backeb.	BV2535	Non-fibrous	Cylindrical
	*C. cornifera* (DC.) Lem.	BV2534	Non-fibrous	Cylindrical
	*C. delaetiana* (Quehl) A. Berger	BV2542	Non-fibrous	Globose
	*C. delicata* L. Bremer	SA1927	Non-fibrous	Cylindrical
	*C. hintoriorum* Dicht & A. Lüthy	BV2539	Non-fibrous	Cylindrical
	*C. macromeris* (Engelmann) Lemaire	BV2600	Non-fibrous	Globose
	*C. pallida* Britton & Rose	SA860	Non-fibrous	Globose
	*C. pseudoechinus* Boed.	BV2543	Non-fibrous	Cylindrical
	*C. ramillosa* Cutak	HSM3775	Non-fibrous	Globose
	*C. retusa* Britton & Rose	SG55	Non-fibrous	Globose
	*Echinocereus cinerascens* (DC.) Lem. subsp. *tulensis*	SA1744	Non-fibrous	Cylindrical
	*E. dasyacanthus* Engelm.	SA2077	Non-fibrous	Cylindrical
	*E. pectinatus* (Scheidw.) Engelm.	SA1918	Non-fibrous	Cylindrical
	*E. pentalophus* (DC.) Lem.	SA1740	Non-fibrous	Cylindrical
	*Mammillaria carnea* Zucc. Ex Pfeiff.	DA241	Non-fibrous	Cylindrical
	*M. dixathocentron* Backeb. Ex Mottram	CPNL133	Non-fibrous	Cylindrical
	*M. magnifica* Buchenau	UG1411	Non-fibrous	Columnar
	*M. mystax* Mart.	DA238	Non-fibrous	Cylindrical
	*Neolloydia conoidea* (DC.) Britton & Rose	BV2595	Non-fibrous	Cylindrical
Opuntioideae	*Cylindropuntia imbricata* (Haw.) F. M. Knuth	TT990	Fibrous	Tree
	*C. kleiniae* (DC.) F. M. Knuth	TT1000	Fibrous	Shrub
	*C. leptocaulis* (DC.) F. M. Knuth	TT994	Fibrous	Shrub
	*Opuntia stenopetala* Lem.	TT997	Fibrous	Shrub
	*O. stricta* (Haw.) Haw.	TT998	Fibrous	Shrub
Pereskioideae	*Leuenbergeria lychnidiflora* (DC.) Lodé	TT967	Fibrous	Tree

The vouchers were deposited in the National Herbarium of Mexico (MEXU). Initial collectors were BV, Balbina Vázquez; SA, Salvador Arias; HSM, Hernando Sánchez-Mejorada; UG, Ulises Guzmán; DA, David Aquino; CPNL, Carmen P. Novoa; TT, Teresa Terrazas. SA, verified species identification.

**Table 2 polymers-14-04840-t002:** Assignment of FTIR absorption bands for the cellulose of Cactaceae species.

Wavenumber (cm^−1^)	Assignments
3000–3600	OH stretching [27]
2900	CH stretching [13,27]
1430	CH_2_ symmetric bending (crystalline and amorphous cellulose) [13,26,27]
1372	C-H and C-O bending vibration bonds [27]
1336	C-O-H in-plane bending (amorphous cellulose) [27]
1315	CH_2_ wagging vibration (crystalline cellulose) [26]
1163	C-O-C asymmetrical stretching [36]
893	Out-of-plane asymmetrical stretching of cellulose ring [13]
670	C-O-H out-of-plane stretching [37]

**Table 3 polymers-14-04840-t003:** Kruskal–Wallis analysis for the crystalline indexes and hydrogen bond index.

Crystalline Indexes	χ-Square	Df	Significance
TCI	65.81053	24	9.25 × 10^−6^
LOI	52.00702	24	7.81 × 10^−4^
HBI	61.89333	24	3.44 × 10^−5^

**Table 4 polymers-14-04840-t004:** Crystallinity indexes with standard deviations, to determine the crystalline cellulose of the 25 Cactaceae species.

Species	TCI (A1370/A2900)	LOI (A1430/A893)	HBI (A3400/A1315)
*C. pallida*	1.118 ± 0.019 a	0.503 ± 0.007 a	1.365 ± 0.032 a
*C. clavata*	1.126 ± 0.012 ab	0.507 ± 0.014 ab	1.301 ± 0.048 ab
*C. delaetiana*	1.263 ± 0.020 abcd	0.515 ± 0.020 ab	1.148 ± 0.018 ab
*C. delicata*	1.227 ± 0.021 abcd	0.523 ± 0.010 ab	1.231 ± 0.033 ab
*C. hintoriorum*	1.273 ± 0.022 abcd	0.561 ± 0.009 ab	1.131 ± 0.024 ab
*C. macromeris*	1.291 ± 0.023 abcd	0.565 ± 0.009 ab	1.131 ± 0.024 ab
*C. pseudoechinus*	1.197 ± 0.030 abcd	0.540 ± 0.009 ab	1.287 ± 0.030 ab
*E. cinerascens*	1.198 ± 0.026 abcd	0.538 ± 0.011 ab	1.339 ± 0.037 ab
*E. dasyacanthus*	1.234 ± 0.039 abcd	0.535 ± 0.014 ab	1.202 ± 0.034 ab
*E. pentalophus*	1.165 ± 0.018 abcd	0.540 ± 0.012 ab	1.270 ± 0.028 ab
*M. dixathocentron*	1.243 ± 0.021 abcd	0.513 ± 0.012 ab	1.279 ± 0.026 ab
*M. magnifica*	1.219 ± 0.025 abcd	0.532 ± 0.012 ab	1.289 ± 0.025 ab
*M. mystax*	1.179 ± 0.028 abcd	0.511 ± 0.013 ab	1.252 ± 0.017 ab
*N. conoidea*	1.212 ± 0.015 abcd	0.513 ± 0.012 ab	1.280 ± 0.030 ab
*C. imbricata*	1.141 ± 0.021 abcd	0.526 ± 0.013 ab	1.277 ± 0.021 ab
*C. kleiniae*	1.192 ± 0.015 abcd	0.528 ± 0.006 ab	1.293 ± 0.034 ab
*C. leptocaulis*	1.242 ± 0.021 abcd	0.567 ± 0.010 ab	1.250 ± 0.019 ab
*O. stenopetala*	1.299 ± 0.028 abcd	0.535 ± 0.011 ab	1.185 ± 0.042 ab
*O. stricta*	1.287 ± 0.023 abcd	0.541 ± 0.012 ab	1.139 ± 0.035 ab
*L. lychnidiflora*	1.202 ± 0.034 abcd	0.510 ± 0.010 ab	1.270 ± 0.037 ab
*M. carnea*	1.323 ± 0.024 abcd	0.540 ± 0.009 ab	1.097 ± 0.028 ab
*C. ramillosa*	1.362 ± 0.027 abcd	0.472 ± 0.011 a	1.085 ± 0.036 ab
*C. retusa*	1.486 ± 0.039 bcd	0.569 ± 0.017 ab	1.030 ± 0.019 ab
*E. pectinatus*	1.606 ± 0.042 cd	0.647 ± 0.015 b	0.516 ± 0.007 b

Different letters in each column indicate significant differences (*p* < 0.05). Mean ± standard deviation (SD).

## Data Availability

Raw data are available in the Figshare repository: https://doi.org/10.6084/m9.figshare.20264472.v1, https://figshare.com/articles/dataset/Data_from_Cellulose_in_secondary_xylem_of_Cactaceae_crystalline_composition_and_anatomical_distribution/20264472 (accessed on 11 June 2022).

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
