# Peer review of "Cellulose in Secondary Xylem of Cactaceae: Crystalline Composition and Anatomical Distribution"

_polymers, 2022, doi:10.3390/polym14224840_

Round 1

Reviewer 1 Report

Please see the attached document

Author Response

We appreciate your thorough review. All modifications suggested were included. In the pdf you may find in yellow those ones accepted and in comments why we did not accept one change  

Reviewer 2 Report

This manuscript is focused on the characterization of the crystallinity composition and anatomical distribution of cellulose in Cactaceae vascular tissue. This is very interesting topic with the possibility of its potential use in various areas. However, the text has several shortcomings.

1.      Lines 29, 30, etc. In the whole text, hemicellulose should be replaced by hemicelluloses, because there are several types of hemicelluloses in the wood.

2.      Lines 48-49. The authors stated that the importance of studying the crystalline composition of cellulose is due to its use in the paper industry and the production of biofuels. However, they should also specify how much biomass of the studied plant species is available annually for potential utilization in the mentioned types of industry.

3.      Line 74. Samples were extracted with ethanol: benzene mixture. However, benzene is an extremely toxic and carcinogenic compound, thus it is not used for extraction worldwide. I strongly recommend in the future work replace benzene by toluene according to ASTM D1107-21 (2021) Standard Test Method for Ethanol-Toluene Solubility of Wood (West Conshohocken, PA: ASTM International).

4.      Line 84. Please, correct Kûshner-Höffer method to Kürschner-Hoffer method. In addition, I recommend using Seifert's methods for the determination of cellulose: Seifert VK (1956) About a new method for rapid determination of pure cellulose (in German: Uber ein neues Verfahren zur schnell Bestimmung der rein-Cellulose). Das Papier 10(13):301–306. The procedure and comparison of both methods is presented in the article: Tribulová, T.; Kačík, F.; Evtuguin, D.V.; Čabalová, I.; Durkovič, J. The effects of transition metal sulfates on cellulose crystallinity during accelerated ageing of silver fir wood. Cellulose 2019, 26, 2625–2638. https://doi.org/10.1007/s10570-018-2210-8

5.      Lines 91-100. What FTIR technique was used (KBr, ATR...)? The range of wavenumbers (4000-650 cm-1) indicates that ATR-FTIR with diamond was used, but this should be clearly stated in the text.

6.      Line 95. cm-1 – upper index

7.      Line 127. Kitin et al. 2020 – the reference should be according to Polymers journal rules.

8.      Line 171. Table 2. cm-1 – upper index, CH2 – lower index.

9.      Lines 400 and further. References. This part must be carefully checked and corrected. Some journals names are abbreviated in bad way (e.g. Materials – Ref. 26, 27; Cellulose – Ref. 36.), etc.

Author Response

We appreciate your comments. In the attached pdf we give response to your comments 

Round 2

Reviewer 2 Report

Please, correct  HNO3 with lower index (line 88).